# Protective Effects of Euthyroidism Restoration on Mitochondria Function and Quality Control in Cardiac Pathophysiology

**DOI:** 10.3390/ijms20143377

**Published:** 2019-07-10

**Authors:** Francesca Forini, Giuseppina Nicolini, Claudia Kusmic, Giorgio Iervasi

**Affiliations:** Institute of Clinical Physiology, CNR, via G.Moruzzi1, 56124 Pisa, Italy

**Keywords:** mitochondrial quality control, thyroid hormone homeostasis, oxidative stress, calcium handling, inflammation, cardiac disease

## Abstract

Mitochondrial dysfunctions are major contributors to heart disease onset and progression. Under ischemic injuries or cardiac overload, mitochondrial-derived oxidative stress, Ca^2+^ dis-homeostasis, and inflammation initiate cross-talking vicious cycles leading to defects of mitochondrial DNA, lipids, and proteins, concurrently resulting in fatal energy crisis and cell loss. Blunting such noxious stimuli and preserving mitochondrial homeostasis are essential to cell survival. In this context, mitochondrial quality control (MQC) represents an expanding research topic and therapeutic target in the field of cardiac physiology. MQC is a multi-tier surveillance system operating at the protein, organelle, and cell level to repair or eliminate damaged mitochondrial components and replace them by biogenesis. Novel evidence highlights the critical role of thyroid hormones (TH) in regulating multiple aspects of MQC, resulting in increased organelle turnover, improved mitochondrial bioenergetics, and the retention of cell function. In the present review, these emerging protective effects are discussed in the context of cardiac ischemia-reperfusion (IR) and heart failure, focusing on MQC as a strategy to blunt the propagation of connected dangerous signaling cascades and limit adverse remodeling. A better understanding of such TH-dependent signaling could provide insights into the development of mitochondria-targeted treatments in patients with cardiac disease.

## 1. Introduction

Mitochondria play pivotal roles within the cardiac tissue. Besides providing the energy for continuous heart beating, mitochondria coordinate other critical specialized functions, such as calcium (Ca^2+^) concentration sensing and Ca^2+^ buffering, metabolite synthesis, and the integration of cell death and survival pathways. As a consequence, the maintenance of a healthy, functional mitochondrial network is of the utmost importance to preserve cell viability and cardiac performance in response to physiological and stress stimuli, especially considering the very limited regenerative capacity of the mature heart. To preserve a high mitochondrial quality and energy reserve, the cell needs to integrate, near concurrently, multiple mechanisms of mitochondrial quality control (MQC) consisting of the recognition and isolation of irreparably damaged mitochondrial components, their targeting to the clearance systems, and their replacement with plenty of functional ones via mitochondrial biogenesis. When these quality check processes are impaired, as in cardiac disorders, mitochondria become more susceptible to danger signals, which predispose to energy crisis and disease progression [1,2]. 

Oxidative stress, altered calcium handling, and inflammation are interconnected noxious stimuli linking mitochondrial dysfunctions to the pathogenesis of cardiac diseases, including ischemia/reperfusion injury (IRI), cardiomyopathy, and heart failure (HF). Reactive oxygen species (ROS) are common by-products of oxidative phosphorylation (OXPHOS), mainly generated at complex I and III of the electron transport chain (ETC) [3]. While low ROS levels are involved in physiological signaling [4,5], excessive ROS production triggers direct oxidative damage to mitochondrial membrane lipids and ETC proteins, with an adverse impact on mitochondrial adenosine triphosphate (ATP) generation and cardiomyocyte survival [6,7]. Uncontrolled ROS production also favors proteotoxicity indirectly by damaging mitochondrial DNA (mtDNA) [7]. Mitochondria possess their own genome that codes for critical subunits of the ETC complexes. The closeness to the source of ROS production, and the absence of protective histones, render mtDNA particularly susceptible to point mutations, deletion, or a decreased copy number, which impairs OXPHOS and energy production, thus accelerating the development and progression of left ventricle (LV) remodeling and HF [7,8]. In addition, mitochondrial-derived ROS (mtROS) also favor excessive fibrosis through the transforming growth factor beta (TGFβ) profibrotic pathway, which in turn increases mtROS, perpetuating the dangerous signaling of oxidative stress [9,10]. Mitochondrial Ca^2+^ overload is another frequent finding in cardiac pathologies, especially in acute IRI, where it amplifies the deterioration of mitochondrial function and the cell loss by increasing ROS formation and opening the mitochondrial permeability transition pore [11]. High ROS levels, Ca^2+^ overload, and mtDNA released by dead cells, trigger a pro-inflammatory response mediated by the activation of the nuclear factor kappa B (NF-kB) and Nucleotide-binding domain leucine-rich repeat pyrin domain containing 3 family pyrin domain containing 3 (NLRP3) inflammasome pathways. The resulting over-production of inflammatory cytokines further exacerbates mitochondrial impairments and adverse remodeling [12,13]. 

Preventing or blunting such connected, mitochondria-dependent, vicious cycles of pathological mechanisms is considered one major goal for effective cardioprotective strategies [14]. 

In this regard, the maintenance of thyroid hormone (TH) homeostasis is of pivotal importance to contrast cardiac noxious stimuli and to improve the recovery of cardiac performance [15,16]. The crucial involvement of TH signaling in the regulation of cardiovascular physiology has been clearly established in the last decades [17,18]. TH enhance the cardiac performance and heart rate, both locally, by influencing myocardial gene expression or protein activity, and indirectly (systemically), by decreasing the peripheral resistance. One predominant mechanism by which the biologically active TH, triiodothyronine (T3), directly affects heart function is by binding to TH nuclear receptor alpha and beta isoforms (THRα and THRβ). Besides this so called genomic action of TH, non-genomic TH activities exist that are not dependent on THR binding and are initiated at the plasma membrane level [19]. Since all these modalities of TH signals are present in the cardiovascular system, it is not surprising that the maintenance of euthyroidism is of crucial importance to preserve cardiac performance, as documented by the opposite cardiovascular alterations observed under hypo- and hyperthyroidism [17]. Besides overt thyroid dysfunctions, even milder alterations of TH homeostasis may adversely impact the heart function [17]. Cardiac stress conditions, including IR, HF, or cardio-surgical procedures, are associated with a reduction of the circulating T3 [19,20,21]. It is now accepted that this condition, known as a low T3 state (LT3S), favors cardiac disease evolution and worsens patient prognosis [19,22,23]. Consistently, increasing clinical and experimental findings indicate that TH replacement may represent a reparative therapeutic intervention for the diseased heart and that mitochondria are the main target of TH-mediated cardioprotection [16,21,24,25,26,27]. T3 has been shown to control mitochondrial respiration and metabolism and to blunt mitochondrial mediated cell death within the diseased myocardium [16,27,28,29,30,31]. So far, however, the metabolic and cardioprotective actions of T3 have been mainly discussed as separate phenomena. Emerging evidence, based on classical monofactorial studies and omic approaches, indicates interdependency between the metabolic action of T3 and the induction of different processes of MQC under physiological and stress conditions. Therefore, the aim of this review is to provide the state-of-the-art of the available data linking the pleiotropic effects of T3 in contrasting oxidative stress, Ca^2+^ handling, and inflammation to the activation of MQC. Understanding how these T3-dependent signals contribute to the maintenance of mitochondrial homeostasis could provide insights into the development of targeted treatments in patients with LT3S. In view of that and to encourage further investigation, novel putative mechanisms of T3-mediated MQC are also discussed.

## 2. TH and Oxidative Stress

Oxidative stress ensues when the ROS production rate overwhelms the cell antioxidant capacity. Mitochondria are both the trigger and main target of oxidative stress, which represents a causal mechanism for both acute and chronic cardiac disease evolution [6,7,32,33]. Reducing ROS production and enhancing the antioxidant defenses are therefore critical steps that can be taken to limit oxidative stress damage, mitochondrial dysfunction, and adverse remodeling. In this context, several lines of evidence indicate that TH are able to modulate the redox balance of the cardiac cells under different stress conditions and by multiple mechanisms (Figure 1) [30,34,35,36,37,38]. As a common finding, TH administration decreases ROS levels and blunted mitochondrial and cellular oxidative damage both in cultured cardiomyocytes exposed to oxidative stress and in models of IR and HF. One important underlying mechanism is the upregulation of cytosolic and mitochondrial antioxidant enzymes. In human cardiac cell lines exposed to simulated IR, T3 administration improved the redox state and inhibited intrinsic cell death by inducing Glutathione peroxidase (Gpx) and Superoxide dismutase (SOD) [37]. In models of acute IR, the maintenance of TH homeostasis resulted in increased mitochondrial levels of Sod1 and Sod2, Aldehyde dehydrogenase 2 (Aldh2), and Peroxiredoxin 2 (Prdx2) and 5 (Prdx5), which was related to increased superoxide clearance, better preserved mitochondrial activity, and more favorable cardiac recovery [15,16]. 

Enhancement of cardiac Nitric oxide synthase (eNOS) expression and NO availability greatly contributes to the TH-dependent antioxidant effect [35,36]. In animal models of LV and RV HF, T3 and T4 activate the AKT/eNOS axis, leading to the increased expression of Peroxisome proliferator activated receptor gamma coactivator 1 alpha (PGC1α) [35,36]. In turn, the latter transcription cofactor is directly implicated in the enhancement of the mitochondrial antioxidant system [39,40]. The AKT/eNOS/Pgc1α antioxidant pathway, initiated at the plasma membrane, is an example of the indirect regulation of gene expression by T3 and T4 (Figure 1).

Another route of indirect transcriptional control of the redox balance is via TH-dependent microRNAs. A recent integrative analysis of T3 differentially-expressed cardiac genes and miRNAs in the post-ischemic setting has identified a network of regulatory circuits that are predicted to inhibit oxidative stress, cell death, and adverse remodeling, while preserving mitochondrial metabolism and integrity [16]. In this scenario, a key role is played by the T3-mediated reduction of the post ischemic levels of mirRNA-31, -155, and -222, which corresponded to the increased expression of their putative targets Sod1 and Sod2 (Figure 1). The dangerous mitochondrial effects of these T3-down-regulated miRNAs have been previously implicated in adverse cardiac dysfunction during ischemic heart disease and HF [41,42,43,44], thus suggesting a novel mechanism for the indirect TH regulation of oxidative stress.

Opening of the mitochondrial ATP-dependent potassium channel (mitoK-ATP) has also been reported as a possible T3-regulated antioxidant mechanism [30]. The activation of mitoKATP under anoxic conditions reduces the ROS concentration and favors the maintenance of mitochondrial membrane potential and ATP production [45]. In cultured cardiomyocytes exposed to pro-oxidant injuries, T3 pre-treatment reduced cell death and preserved mtDNA and ETC function, which was prevented by inhibition of the mitoK-ATP channel opening (Figure1). The exact mechanism underlying the T3-mediated activation of mitoKATP is still elusive; uncovering of the channel molecular identity will help to clarify this important issue.

Collectively, the available data indicate a close connection between the multiple antioxidant effects of TH and the rescue of mitochondrial integrity and bioenergetics. 

## 3. TH Regulation of Mitochondrial Calcium

Ca^2+^ homeostasis is essential for rhythmic heart contraction/relaxation activity and for coupling energy production to the cardiac demand. Mitochondria are major integrators of Ca^2+^ signaling involved in metabolism and cell fate [46]. Under a physiological condition, Ca^2+^ fluctuations stimulate energy production and cell survival. Several metabolic enzymes of the tricarboxylic acid (TCA) cycle are Ca^2+^-dependent, including pyruvate dehydrogenase, isocitrate dehydrogenase, and α-ketoglutarate dehydrogenase. Ca^2+^ also regulates mitochondrial aspartate/glutammate carriers, as well as proteins involved in ROS scavenging [47,48]. Under acute or chronic cardiac stress, prolonged mitochondrial Ca^2+^ overload in acute ischemia and HF contributes to mitochondrial dysfunction, leading to the activation of apoptotic and necrotic cell death pathways [11]. These actions underlay the critical importance of preserving the Ca^2+^ dynamic [46]. 

Ca^2+^ handling and mechano-energetic coupling rely on the integrated activity of the mitochondria and the main intracellular Ca^2+^ store, the sarcoplasmic reticulum (SR). The magnitude of the Ca^2+^ released by the SR at each systole is dependent on the enzymatic activity of SR Ca^2+^ ATPase (Serca2a), which facilitates Ca^2+^ re-uptake into the SR in the diastolic phase. A variety of cardiovascular diseases, including myocardial infarction, cardiac hypertrophy, arrhythmias, IR, and HF, are characterized by the dysfunction of Serca2a levels and activity, resulting in cytosolic and mitochondrial Ca^2+^ overload. By contrast, Serca overexpression rescues the efficiency of coupling between cardiac work and the TCA cycle and improved contractility in post-ischemic heart diseases [49,50,51,52,53]. Therefore, Serca2a is considered a potent therapeutic target for cardiovascular disease [49]. 

TH have widely been reported to favor Ca^2+^ re-uptake within the SR, both in vivo and in vitro (Figure 2). This effect is achieved by transcriptional up-regulation of Serca2a and by down-regulation of its specific inhibitor, phospholamban [18,54,55,56,57]. The LT3S observed in IR and HF may thus contribute to worsening Ca^2+^ signaling and cardiac performance by favoring a reduction in Serca2a activity. Accordingly, T3 replacement in patients with post-IR LowT3S mitigated regional contractile dysfunctions [21]. 

Besides this crucial function, novel data suggest the potential involvement of T3-modulated miRNAs in controlling SR Ca^2+^ release and mitochondrial Ca^2+^ uptake (Figure 2). One main effector of SR Ca^2+^ release in the cardiomyocytes is the Inositol 1,4,5-trisphosphate receptors (IP3R). Uncontrolled IP3R activity is implicated in triggering Ca^2+^-dependent cardiac hypertrophy and cell death [58,59]. In a heart disease model, mir-133 has been shown to restrain Ca^2+^ signaling alterations by targeting IP3R [58]. The major player in the control of mitochondrial Ca^2+^ uptake is the recently identified mitochondrial Ca^2+^ uniporter (MCU) [60]. Under a physiological condition, MCU activity couples the contractile Ca^2+^ cycling with the mitochondrial energy output [61]. Conversely, under pathological conditions, MCU is the principal trigger of mitochondrial Ca^2+^ overload. Consistently, MCU suppression afforded cardioprotective effects in models of acute IR and HF [61,62]. MCU is a validated target of mir-1 and lowering of the MCU content by mir-1 up-regulation is regarded as cardioprotective against mitochondrial Ca^2+^ overload in a cardiac overload stress condition [63]. T3 treatment in the post-IR setting increased the myocardial level of both mir-133 and mir-1 that were down-regulated by the IR event [16,64]. Based on the above data, the T3 regulatory function of these myo-miRNAs may have a beneficial impact on the maintenance of Ca^2+^ homeostasis and heart function by blunting the iper-activation of IP3R and MCU [16,64]. 

Finally, the coupled regulation of mitochondrial metabolism and cardiac performance requires the presence of high Ca^2+^ microdomains that are generated at sites of juxtaposition between the mitochondria and SR. Several lines of evidence point at a key role of mitofusin 2 (Mfn2) in the formation of these so-called mitochondrial-associated membranes (MAM) [65,66]. 

Mfn2 ablation resulted in altered mitochondrial Ca^2+^ handling and impaired heart bioenergetics [67]. In this context, post-ischemic LT3S was associated with a decreased expression of Mfn2 and altered mitochondrial function, which was reverted by LT3S correction with T3 replacement [16]. These findings are suggestive of a putative role of T3 in the control of MAM formation, which deserves to be investigated in depth. 

## 4. TH Regulation of Cardiac Inflammation

The cardiac inflammatory process is directly related to adverse LV remodeling and HF progression. Mitochondrial dysfunction and mtROS are the main triggers of the myocardial inflammatory response in acute IR and in chronic cardiac disease [12,67]. Therefore, mitochondria-targeted antioxidant interventions are expected to reduce inflammation markers as well. Accordingly, in several independent studies, T3 replacement in the post-cardiac ischemia setting alleviated the inflammatory response in connection with decreased ROS production, improved cardiac recovery, and the rescue of mitochondrial function (Figure 3) [16,67]. In particular, De Castro and coworkers demonstrated a critical role for T3 and T4 treatment in blunting the activation of the Toll-like receptor 4 (Tlr4)/NF-kB pro-inflammatory axis at two weeks following myocardial ischemia [68]. These findings were correlated with cardiac functional improvement and the decreased activation of xanthine oxidase, an important source of ROS in the ischemic heart. The Tlr4/NF-kB pathway has been previously shown to enhance the production of cytokines and chemokines in the infarcted heart, promoting a pro-inflammatory cascade leading to fibrosis and pathological cardiac remodeling [69]. By contrast, inhibition of the NF-kB pathway after myocardial infarction mitigated adverse remodeling while reducing TGFβ signaling and blunting extracellular matrix derangement [70]. Consistently, in a transcriptional array study, we found that a timely T3 replacement following IR down-regulated the early expression of NF-kB, along with an array of pro-inflammatory mediators, such as C-X-C motif chemokin receptor 4 (Cxcr4), Integrin-linked kinase (Ilk), Serpine1, Fas ligang (Faslg), and Hepatocyte growth factor (Hgf) (Figure 3) [16]. To suggest a possible underlying mechanism, an in silico analysis revealed that all these factors are a common predicted target of a set of T3-upregulated microRNAs, including mir-144 and miRNAs of the 30 and 133 families [16]. These findings were paralleled by the decreased expression of a signature of genes involved in pathological processes, including intrinsic cell death, mitochondrial dysfunction, mitochondrial-mediated oxidative stress, and the TGFβ pro-fibrotic cascade [16]. Along the same line, in models of post-IR HF, chronic T3 replacement prompted long-term protection against heart dysfunction, which was correlated to an improved expression of markers of mitochondrial function and down-regulation of profibrotic mediators [23,71]. Interestingly, such chronic T3 replacement also blunted the expression of Beta-secretase 1 (Bace 1). This factor is a well-known proinflammatory trigger in the brain cortex implicated in degenerative disease. Bace1 brain content was also elevated in a model of post-ischemic congestive HF, thus suggesting that neuroinflammation may be secondary to myocardial infarction [72]. To validate this interpretation, a recent study reported that post-ischemic myocardial inflammation triggered neuroinflammation in the long run via a mitochondria-derived proinflammatory mediator that was firstly produced within the infarcted myocardium [73]. Based on this premise, it can be hypothesized that the mitochondria-targeted anti-oxidant and anti-inflammatory action of T3 may thus blunt the interconnected heart and brain dysfunctions. Although highly speculative, this is a critical issue worthy of future investigation. 

Collectively, the above studies suggest a crucial protective role of T3 against the interdependent noxious processes linking mitochondrial impairments to cardiac stress, inflammation, and adverse remodeling.

## 5. TH Regulation of Mitochondrial Repair Systems

When ROS scavenging is insufficient to prevent oxidative damage, the repair of misfolded proteins and damaged mtDNA is one further important T3-dependent process of MQC implicated in cardioprotection (Figure 4). Protein folding and repair is achieved via specialized molecules termed chaperones that include the heat shock proteins (HSPs) [74,75]. This surveillance system is of critical importance for cardiac proteostasis and an impairment in its components worsens cardiac diseases [76]. The cardioprotective action of the large HSPs 70 and 90 against IRI and HF has been widely demonstrated by using transgenic animals and isolated cardiac myocyte-derived cells [74,75]. In particular, HSP70 exerts multiple functions to maintain mitochondrial proteostasis involved in de novo folding of nascent polypeptides, the repair of misfolded proteins, mitochondrial import of precursor proteins, and chaperone-mediated protein degradation (see below). Additionally, small HSPs such as α-crystallin B (CryaB) and HSP27 have been shown to protect the cardiomyocytes from proteotoxicity and irreversible damage in in vitro and in vivo models of IR or in human atrial fibrillation [77,78,79]. Recent mitochondrial proteomic profiling showed that retention of T3 homeostasis following cardiac IR resulted in a higher content of both large HSPs 70 and 90, and small HSP27 and CryaB (Figure 4) [15]. These findings were paralleled by better preserved mitochondrial function and cardiac performance, thus implying the involvement of preserved proteostasis in the cardioprotective effect of T3. 

Another functional role of HSP70 and 90 chaperons is to keep misfolded proteins in solution for protein degradation when refolding is not an option [80]. Chaperon-presented proteins are mainly degraded by the ubiquitin proteasome system (UPS), a proteolitic system specialized to remove ubiquitynated damaged proteins (Figure 4) [76]. The Forkhead box O (Foxo) transcription factors play a pivotal role in orchestrating the expression of genes involved in UPS in several tissues, including the heart [81,82]. Emerging evidence from different organs indicates a critical function of TH as regulators of Foxo gene expression, which has a profound impact on tissue homeostasis, protein quality control, and mitochondrial activity [83,84]. In line with these notions, T3 has been shown to promote the up-regulation of the UPS components in cardiac tissue as a strategy to maintain mitochondrial proteostasis in the presence of elevated oxidative metabolism [85].

Besides removing damaged proteins, the UPS pathway may serve to control the cellular level of proteins involved in cell growth and apoptosis. In the ubiquitinization process, the last step of conjugation to ubiquitin is catalyzed by enzymes of the E3 ubiquitin ligase family. In the cardiac tissue, the Mouse double minute 2 homolog (MDM2) is the specific E3 ubiquitin ligase for p53 degradation and is essential to preserving redox homeostasis and mitochondrial bioenergetics (Figure 4) [86]. Reduced myocardial levels of MDM2 and increased p53 protein contents have been observed in both cardiac ischemia and pressure overload in conjunction with oxidative stress, broad mitochondrial deficiency, and apoptosis [86]. Therefore, keeping the p53 level low via MDM2 ubiquitination may represent a cardioprotective MQC strategy to ameliorate mitochondrial function under cardiac stress conditions. A recent microarray approach identified MDM2 as a T3 responsive gene necessary for the physiological growth of the muscle cell [87]. Although the effect of TH on the myocardial MDM2/p53 axis has never been analyzed, recent papers described the involvement of T3 in down-regulating p53 under cardiac IR, thus supporting such a working hypothesis [16,31].

Maintenance of mtDNA integrity is another crucial process that can be employed to preserve the functional activity of the mtDNA-encoded proteins. Enhancement of the mtDNA repair machinery is considered a novel therapeutic target to prevent acquired mtDNA damage and mutations induced by noxious stimuli in the evolution of HF [8]. In a model of IR, early recovery of euthyroidism elicited an increased mitochondrial level of proteins implicated in mtDNA repair that is: 40s ribosomal protein S3 (rpS3), the O-acetyl-ADP-ribose deacetylase (Macrod1), and Aconitase (Aco) (Figure 4). Upon oxidative stress, RpS3 is translocated to the mitochondrial compartment via HSP70 and reduces the ROS-mediated mtDNA damage [88]. As RpS3, Macrod1 harbors a mitochondrial targeting sequence and has been suggested to exert a reparative action towards ADP-ribosylated mtDNA adducts [89]. In several cell types, the mitochondrial TCA enzyme Aco has been shown to prevent oxidant-mediated mtDNA damage and apoptosis by coupling TCA cycle progression to mtDNA packaging [90,91]. Therefore, the maintenance of mitochondrial protein homeostasis and mtDNA integrity by T3 replacement may be a further cardioprotective mechanism linking improved MQC to better preserved cell function and cardiac performance either in physiological conditions or following IR stress.

## 6. TH Regulation of Mitochondrial Clearance

When mitochondrial damages overwhelm antioxidant defenses and the molecular repair capacity, mitochondrial selective degradation (mitophagy) is the extreme process of MQC that occurs to avoid the propagation of dangerous pathways [92]. In acute and chronic cardiac disease, mitophagy fosters the sequestration and degradation of damaged mitochondria that might otherwise trigger cardiomyocyte dysfunctions and cell death [92]. In addition, mitophagy prevents the activation of NLRP3 inflammasome by oxidized mitochondrial phospholipids and mtDNA [92]. Accordingly, impaired mitophagic flux and the accumulation of dysfunctional mitochondria have been linked to HF progression, thus indicating the therapeutic potential of targeting mitophagy in heart diseases [1,93,94]. The trigger for the selective elimination of dysfunctional organelles is prolonged, irreversible inner mitochondrial membrane depolarization resulting from a loss of oxidative phosphorylation capacity. During the mitophagic process, dysfunctional mitochondria are recognized and progressively engulfed into double membrane vesicles, called autophagosomes, that are coated with the marker light chain 3 (LC3). Then, fusion of the autophagosomes with lysosomes determines vesicle breakdown and degradation [94]. In various animal models TH have been shown to stimulate virtually every step of the mitophagic flux (Figure 5). Emerging evidence indicates that TH couple mitophagy and biogenesis to enhance mitochondrial turnover, thus protecting the cells against oxidative damage derived from both physiological oxidative metabolism or noxious stimuli [83,95,96,97,98,99]. The underlying mechanism is the simultaneous up-regulation of Pgc1α and a plethora of critical factors involved in the formation of the autophagosome. Among them, the Unc-51-like autophagy activating kinase1 (ULK1) and Beclin1 (Becn1) are involved in initiation and elongation of the phagophore. Then, the recruitment of LC3 is necessary for the recognition of the damaged mitochondria. One of the recognition mechanisms occurs via mitochondrial surface proteins such as BCL2 Interacting Protein 3 (Bnip3) and FUN14 domain-containing protein 1 (Fundc1) that act as receptors for L3 [94]. Another mechanism is via the PTEN-induced kinase 1(Pink1)/Parkin (Park) axis. In intact mitochondria, Pink1 is imported within the mitochondria matrix and degraded. In injured mitochondria, inner membrane depolarization enables Pink1-stabilization at the outer mitochondrial membrane, which allows Mfn2 phosphorylation and recruitment of Park. Park then ubiquitinates proteins at the mitochondrial surface where the p62 adaptors form a bridge connecting ubiquitin to LC3 (Figure 5) [94]. In skeletal muscle, T3 increased the overall mitophagic flux by enhancing the expression and activity of mitophagic executors such as Ulk1, Bnip3, LC3, and p62, as well as by up-regulating the upstream inducers of autophagy Foxo1 and Foxo3. Interestingly, inhibition of the autophagic program repressed mitochondrial activity and biogenesis induced by T3, thus suggesting that T3-mediated mitophagy contributes to maintaining MQC and mitochondrial function [83]. The same connection was also found in T3-treated brown adipose tissue and liver [95,96,97,98,99,100]. In the above models, simultaneous transcriptional induction of both Pgc1α and of Ulk1, Fundc1, and LC3 was necessary to enhance mitochondrial clearance and to preserve mitochondria respiration and lipid oxidation [95,96,97,98]. Furthermore, T3 has been demonstrated to induce protective mitochondrial clearance against infective or ischemic liver injuries [84,99,100]. In those contexts, T3 favored mitochondrial turnover via the up-regulation of Beclin1, Pink/Park, and LC3, which was coupled to increased mitochondrial biogenesis and was essential to reducing ROS-inflicted cell damage and death [84,99,100]. Recent findings also strongly suggest a role of T3 in the modulation of mitochondrial clearance in cardiac disease conditions [16,71]. In a system biology study, a post-IR LT3S was associated with impaired mitochondrial activity in association with a reduced expression of key markers of mitochondrial biogenesis, mitophagy, and fusion, including Pgc1α, Bnip3, Mfn1, and Mfn2 [16]. These alterations were mitigated by early and short-term T3 replacement and were associated with a better recovery of cardiac function [16]. In stressed cardiomyocytes, Mfn2 contributes to maintaining MQC by promoting mitochondrial fusion as well as park-dependent protective mitophagy against ROS-induced mitochondrial damages and cell death [101]. Mfn2-mediated tethering of SR and mitochondria is crucial for the activation of protective mitophagy in the IR heart [14]. In light of these data, it can be speculated that the enhancement of mitochondrial fusion and mitophagy contributes to the protective effects of T3 in post-IR wound healing repair. To reinforce this interpretation, in a recent study, chronic post-IR T3 replacement blunted adverse cardiac remodeling and atrial arrhythmias, which was accompanied by the restoration of gene expression involved in mitochondrial biogenesis, ETC, and mitophagy including Tfam, Beclin1, and LC3 [71]. 

Recently several miRNAs have been identified as critical modulators of the cardiac autophagic process (Figure 5) [94]. Among them, mir-144 exerts protective effects against cardiac ischemic injury by targeting the Mammalian target of rapamycin (mTOR), a well-known inhibitor of autophagy and mitophagy [102]. On the other hand, mir-222 targets p27, an inhibitor of mTOR, thus restraining the autophagic flux and favoring HF evolution [43]. We showed that post IR T3 replacement enhances the myocardial expression of mir-144, while inhibits mir-222 levels [16]. These findings further strengthen the critical involvement of T3 in the autophagic removal of mitochondria under stress conditions. 

## 7. TH Regulation of Mitochondrial Biogenesis and Protein Import Machinery

The maintenance of a functional mitochondrial network requires degraded mitochondria to be replaced with new mitochondrial components synthesized through mitochondrial biogenesis.

Most of the mitochondrial proteome is encoded by the nuclear genome, while the mitochondrial DNA encodes for 13 essential polypeptides of the ETC complexes. A tight coordination of the two genomes is necessary to ensure the proper assembly of all the mitochondrial components. The master regulator of the nuclear mitochondrial crosstalk is PGC-1α, the co-activator that interacts with a variety of transition factors involved in mitochondrial biogenesis, bioenergetics, and antioxidant activity [103]. Among the PGC-1α targets, the nuclear-encoded mitochondrial transcription factor A (Tfam) is responsible for the replication and repair of the mitochondrial genome and for the transcription of mtDNA-encoded genes [104]. Several lines of evidence have established a clear connection between TH induction of mitochondrial activity and up-regulation of the entire process of mitochondrial biogenesis [105,106,107,108,109]. One main mechanism is via T3 binding to TH receptors located in both the nuclear and mitochondrial compartments [105,106,107,108,109]. The key role of TH in the up-regulation of PGC-1α and Tfam has been described in previous reviews [27,105,106,107,108,109]; here, we focus on the regulation of the mitochondrial protein import system by T3. 

The vast majority of the nuclear-encoded mitochondrial proteins are produced as precursor polypeptides harboring a mitochondrial targeting sequence which allows the correct final cellular location. The precursor proteins are recognized by the receptors of the translocase machinery situated at the outer mitochondrial membrane (TOM20 complex) and internalized by members of the inner mitochondrial membrane traslocase mechinery (TIM23 complex). In this import process, HSP70 associates with the TIM-23 complex and mediates pre-protein movement through its ATP-hydrolyzing activity (Figure 5) [110]. 

Early studies in hypothyroid and hyperthyroid models and isolated cardiac mitochondria highlighted the involvement of T3 in the up-regulation of several components of the mitochondrial import complexes, including Tom20, Tim23, Tim44, and HSP70 [111,112,113,114,115]. Besides increasing the gene expression and protein levels of these import machinery components, T3 also favored the mitochondrial translocation of HSP70 and the import of the mitochondrial enzymes malate dehydrogenase and ornithine carboxytransferase. 

According to recent data, T3 may also be involved in mitochondrial protein import in ischemic cardiac disease [15,16]. A post-IR LT3S resulted in reduced mitochondrial levels of HSP70 and Metaxin2 (Mtx2), as well as in a decreased expression of Tim17a and Tim8b, which was prevented by the maintenance of T3 homeostasis [15,16]. Mtx2 is situated at the outer mitochondrial membrane. It is involved in protein import independently of the TOM20 system and possesses a glutathione-tranferase domain to detoxify reactive compounds by conjugation to glutathione [114]. Tim17 is situated at the core of the TIM23 complex, where it plays an essential role in the formation, stabilization, and voltage gating of the translocation channel. Tim8 is part of a complex that becomes essential to driving the import of the core component of the TIM23 complex (Tim23) and to preventing its retrograde translocation in conditions of reduced inner membrane potential [115]. Therefore, increasing the mitochondrial level of HSP70, Mtx2, Tim17, and Tim 8 may provide cardioprotection by promoting the import of mitochondrial proteins in conditions of reduced inner membrane potential (Figure 5). 

Collectively, these data suggest that the activation of mitochondrial biogenesis and protein import by T3 may contribute to rescuing mitochondrial function by favoring protein turn-over, even under a cardiac stress condition. 

## 8. Conclusions

Under cardiovascular disease conditions, such as IR and HF, multiple pathways of MQC become exhausted, thus exacerbating cell death and cardiac dysfunction. In this scenario, a therapeutic strategy aimed at enhancing the multilayer process of MQC is expected to improve the energy supply, leading to a better preservation of contractile force and inhibition of the fibrosis–cell death axis. The available data indicate that the restoration of euthyroidism, in the setting of post-IR LT3S, increases myocardial resilience to stress stimuli in association with the retrieval of preventive and reparative pathways that are relevant to cardiomyocytes mitochondrial bioenergetics and QC.

On the one hand, T3 prevents the excessive activation of mitochondrial dangerous cascades by enhancing anti-oxidant and anti-inflammatory effectors, in conjunction with better preserved Ca^2+^ handling. On the other hand, T3 stimulates processes that augment MQC to repair established damage to mtDNA, mitochondrial proteins, and lipids, thus rescuing cell bioenergetics. These actions are achieved at the transcriptional, post-transcriptional, and plasma membrane level to orchestrate multifunctional genes, concurrently controlling multiple pathways of mitochondrial metabolism and MQC. The T3-dependent regulation of Pgc1α, HSP70, Mfn2, and miomiRNAs, clearly exemplifies this multifaceted, integrative strategy of gene expression control. Pgc1α is implicated in mitochondrial biogenesis and metabolism, but also in antioxidant activity and mitophagy. HSP70 plays a crucial role in the mitochondrial import of ETC proteins and matrix enzymes, as well as in the folding of nascent pre-polipeptides and in the refolding or degradation of damaged proteins. Mfn2 is involved in protective mitochondrial fusion, mitophagy, and MAM formation. Similarly, T3-regulated myo-miRNAs control critical aspects of redox balance, Ca^2+^ handling, and the clearance of irreparably damaged mitochondria. This redundant surveillance system is pivotal under physiological conditions to preserve MQC in front of increased OXPHOS induced by TH and is even more important in a pathological condition to blunt the propagation of dangerous signaling while maintaining mitochondrial function and prevent adverse remodeling. 

So far, the vast majority of investigations aimed at identifying TH targets have exploited indirect approaches, mainly based on a comparative analysis of RNA or protein abundance [116]. Further studies on myocardial chromatine occupancy by liganded and unliganded THR are necessary to clearly distinguish between the direct and indirect influence of TH on cardiac gene transcription. However, the available data on T3 cardiac effects should provide helpful guidance to optimize the benefit to risk profile when considering LT3S correction. To be effective against adverse remodeling, euthyroidism reconstitution should be performed in the early phase of the post-IR wound healing process, which is a time when the activation of noxious mitochondrial regulatory networks has not yet produced permanent effects and may be prevented or blunted by T3 replacement. Accordingly, TH administered early after IR enhanced cardiac function and prevented left ventricle remodeling [27,30,31,64]. By contrast, T3 treatment started one week after IR improved heart performance without reversing cardiac remodeling, which may be the result of systemic rather than local myocardial effects [117]. Finally sex-specific differences in the long-term cardiac responsiveness to T3 have been reported in post-acute myocardial infarction patients, which should be taken into account for a better personalization of the TH replacement strategy [118].

In conclusion, increasing experimental data indicate the mitochondrion as a main target of TH-mediated cardioprotection. An approach aimed at euthyroidism restoration by restoring the free T3 plasma levels without major systemic alterations may represent the best strategy to limit the evolution of HF while avoiding the undesirable hypermetabolic and pro-oxidant effects of higher TH doses. 

## Figures and Tables

**Figure 1 ijms-20-03377-f001:**
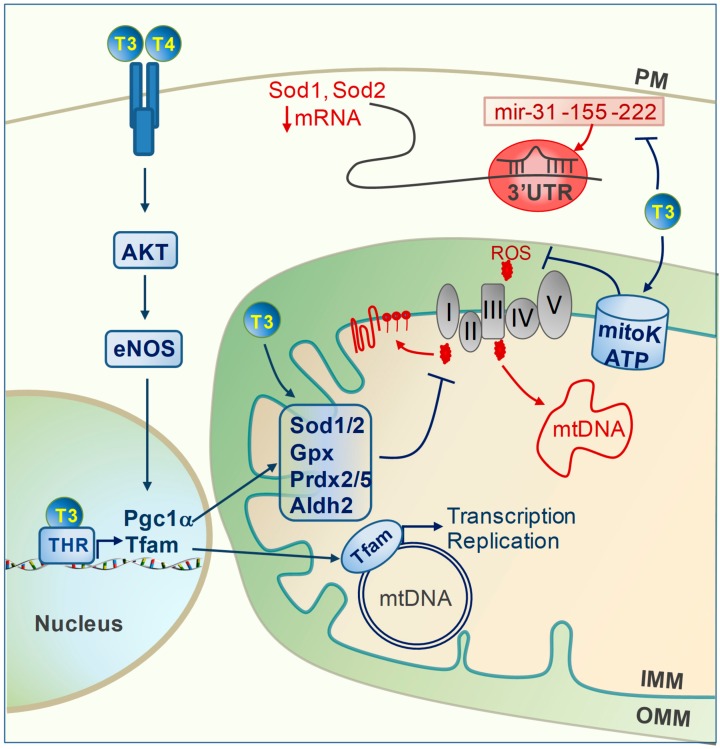
Antioxidant effect of thyroid hormones (TH). Reactive oxygen species (ROS) produced at electron transport chain (ETC) complex I and III can damage mitochondrial DNA, lipids, and proteins. T3 and T4 decrease reactive oxygen species (ROS) production by multiple mechanisms: (1) membrane-initiated pathways as for the AKT/eNOS axis; (2) transcriptional regulation of Pgc1α and Tfam; (3) post-transcriptional regulation via inhibition of mir-31, -155, and -222; (4) activation of the mitoK-ATP protective channel. Effectors modulated by TH are marked in blue. (AKT: protein kinase B; Aldh2:aldehyde dehydrogenase 2; eNOS: nitric oxide synthase; IMM: inner mitochondrial membrane; Gpx; Glutathione peroxidase mtDNA: mitochondrial DNA; OMM: outer mitochondrial membrane; Pgc1α: PPARG coactivator 1 alpha; PM: plasma membrane; Prdx2/5: peroxiredoxin 2/5; Sod1/2: superoxide dismutase1/2; Tfam: mitochondrial transcription factor A; THR: thyroid hormone receptor).

**Figure 2 ijms-20-03377-f002:**
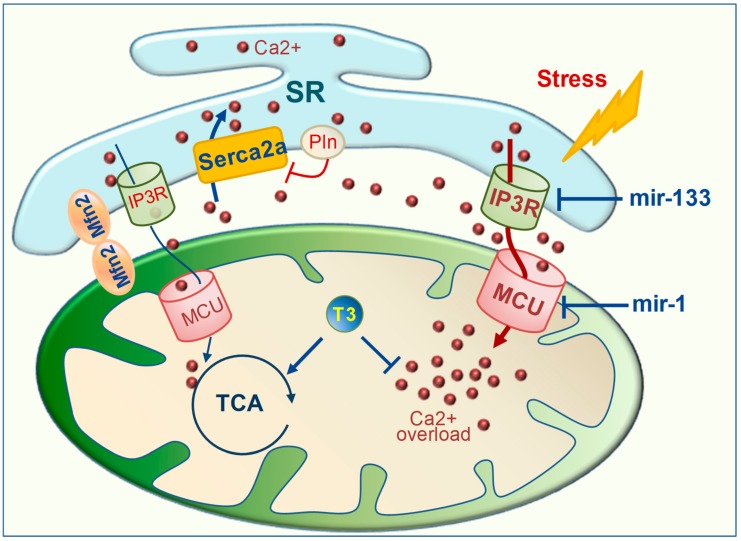
Effect of T3 on Ca^2+^ handling. Under stress conditions, Ca^2+^ dis-homeostasis is determined by a reduced function of the Serca2a channel and increased activity of the Serca2a inhibitor Pln. Excessive activation of IP3R and MCU channels contributes to mitochondrial Ca^2+^ overload, mitochondrial dysfunction, and cell death. T3 limits mitochondrial Ca^2+^ overload and favors energy production through several mechanisms: (1) promoting Ca^2+^ uptake within the SR by up-regulating the Serca2a channel and down-regulating Pln; (2) enhancing the expression level of mir-1 and mir-133, established suppressors of IP3R and MCU, respectively; (3) stimulating the activation of the Ca^2+^ dependent TCA enzymes by Mfn2-dependent tethering of the mitochondrial membrane and SR, thus prompting the formation of Ca^2+^ microdomains. (IP3R: inositol 1,4,5-trisphosphate receptors; MCU: mitochondrial calcium uniporter; Mfn2: mitofusin2; Pln; phospholambam; Serca2a: SR Ca^2+^ ATPase 2a; SR: sarcoplasmic reticulum; TCA: tricarboxylic acid cycle).

**Figure 3 ijms-20-03377-f003:**
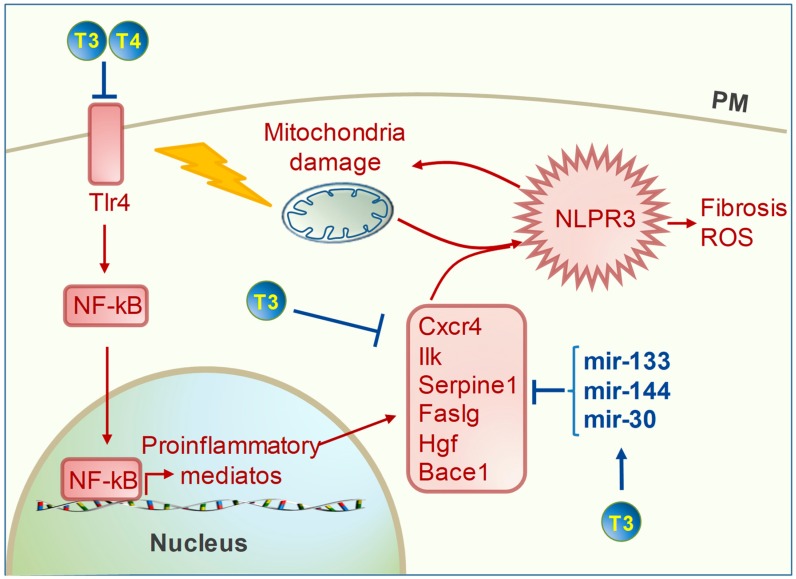
Anti-inflammatory effect of thyroid hormones (TH). Excessive reactive oxygen species (ROS) formation and Ca^2+^ overload lead to mitochondrial dysfunction and activation of the inflammasome NLPR3, which exacerbates mitochondrial impairment and ROS production and activates the fibrotic response. TH limits the inflammasome formation by: (1) blunting the pro-inflammatory Tlr4/NF-kB axis; (2) down-regulating the expression levels of an array of pro-inflammatory mediators; (3) up-regulating the expression levels of mir-133, -144, and -30, predicted suppressors of such pro-inflammatory mediators. (Bace 1: beta secretase 1; Cxcr4: C-X-C motif chemokin receptor 4; Faslg: fas ligand; Hgf: hepatocyte growth factor; Ilk: integrin-linked kinase; NF-kB: nuclear factor kappa B; NLPR3: inflammasome; Tlr4: toll like receptor 4).

**Figure 4 ijms-20-03377-f004:**
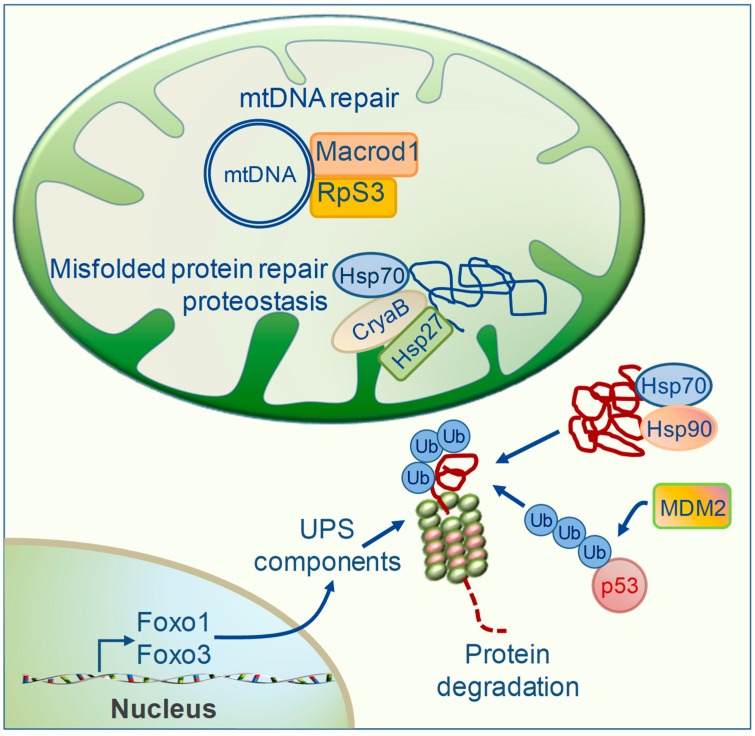
Effects of thyroid hormones (TH) on mitochondrial protein repair and proteostasis. Accumulation of damaged mitochondrial proteins and mutated mtDNA favors mitochondrial impairments and cell dysfunction. TH acts at multiple levels to avoid such dangerous effects. (1) T3 favors the repair of misfolded proteins or their chaperone-mediated degradation by enhancing the protein levels of small (HSP27, CryaB) and large (HSP 70 and 90) heat shock proteins; (2) T3 and T4 up-regulate Foxo1 and Foxo3, transcription factors that orchestrate the expression of components of the ubiquitin protesome system (UPS) for protein degradation; (3) T3 may favor the ubiquitination of p53 under cardiac stress conditions via the MDM2 pathway; (4) T3 favors the mitochondrial accumulation of DNA repair systems such as Macrod1 and Rps3. (CryaB: α-crystallin B; HSP: heat shock protein; Foxo1/3: forkhead box O 1/3; Macrod1: Mono-ADP Ribosylhydrolase; Mdm2: mouse double minute 2 homolog; mtDNA: mitochondrial DNA; RpS3: 40s ribosomal protein S3; ub: ubiquitin).

**Figure 5 ijms-20-03377-f005:**
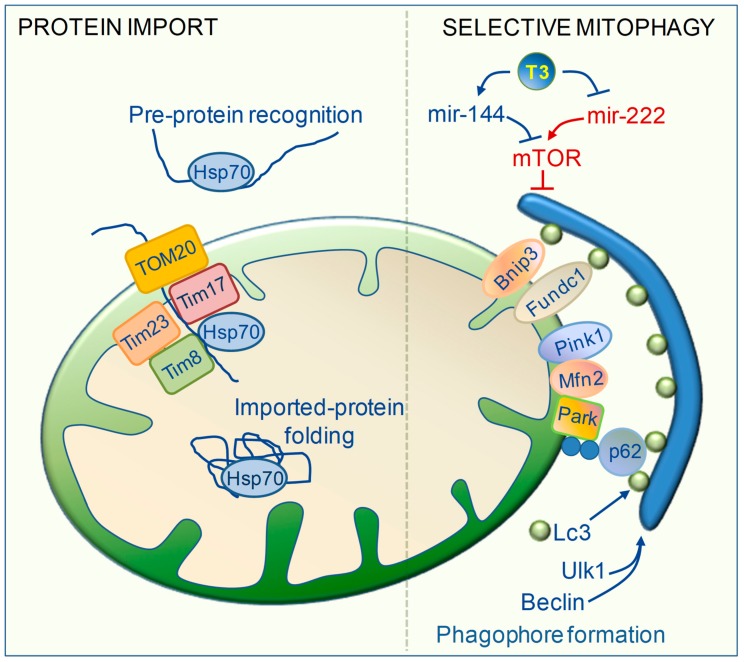
Effect of T3 on mitochondrial protein import and mitochondrial clearance. On the left, T3 enhances mitochondrial protein import by up-regulating the expression and protein levels of translocases of the outer (Tom) and inner mitochondrial membrane (Tim). Also, T3 increases the activity of Hsp70 that is involved in recognition of the mitochondrial pre-proteins, sliding of pre-proteins within translocase complexes and folding the imported pre-proteins. On the right, T3 favors the selective degradation of irreparably damaged mitochondria via mitophagy. (1) T3 up-regulates a set of molecules involved in phagophore formation and elongation including Lc3, beclin and Ulk1; (2) T3 enhances Park-dependent and independent recognition of damaged mitochondria by the Lc3 component of the phagophore; (3) T3 regulates mir-144 and mir-222 thus blunting the inhibitory effect of the mTOR pathway on the mitophagy program. (Bnip3: BCL2 interacting protein 3; Fundc1: FUN14 domain-containing protein 1; Hsp70: heat shock protein 70; Lc3: light chain3; Mfn2: mitofusin2; mTOR: mammalian target of rapamycin; p62: ubiquitin protein binding p62; Park: parkin; Pink1: PTEN induced kinase 1; Ulk1: unc-51 like autophagy activating kinase1).

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
