# Peer review of "Protective Effects of Euthyroidism Restoration on Mitochondria Function and Quality Control in Cardiac Pathophysiology"

_ijms, 2019, doi:10.3390/ijms20143377_

Reviewer 1 Report

The authors Forini et al are experts in field of thyroid hormones (TH) and cardioprotection. In their review, they nicely summarize the effects of TH on mitochondria and discuss the data in the field with special focus on their own previous data. The review contains helpful Figures, a clear outline that addresses all crucial issues of T3 and mitochondria as well as a clear introduction. However, the discussion is relatively short and should take the advantage e.g. of a detailed discussion of the impact of the reviewed issues on therapeutic decisions, the potential exact/molecular role of TH in cardioprotection and potential strategies to selectively target certain functions of TH in order to provide a save reconstitution of TH.

Special comments:

·      Title: The title is not appropriate since the review is focused on the effects of reconstitution of euthyroidism. The title is too general for the content of the review.

·      Introduction:

-       Physiological functions of TH should be included 

-       Molecular signaling of TH in the cardiovascular system (e.g. THRa and THRb?)

-       The specialization on euthyroidism should be introduced/explained. Why is hyperthyroidism not discussed?

-       Mention/Discuss local versus systemic effects of TH.

·      Discussion:

-       Direct and indirect effect and lack of knowledge with regard to direct target of T3/T4 should be discussed

-       What issues be considered to reconstitute euthyroidism?

-       Sex-related differences are observed in the clinic. This impact should be discussed briefly

·      Figure 1: Role of T4 was not mentioned anywhere

·      Please, check if all abbreviations are explained (e.g. NLPR3 in fig legend 3; THR in fig legend 1,….)

·      Reference 35 is not included in the text.

Author Response

Reviewer #1 (Comments to the authors)

The authors Forini et al are experts in field of thyroid hormones (TH) and cardioprotection. In their review, they nicely summarize the effects of TH on mitochondria and discuss the data in the field with special focus on their own previous data. The review contains helpful Figures, a clear outline that addresses all crucial issues of T3 and mitochondria as well as a clear introduction. However, the discussion is relatively short and should take the advantage e.g. of a detailed discussion of the impact of the reviewed issues on therapeutic decisions, the potential exact/molecular role of TH in cardioprotection and potential strategies to selectively target certain functions of TH in order to provide a save reconstitution of TH. 

R: We are grateful to the reviewer for his/her constructive comments and     suggestions.

To take into account these general indications, the conclusion section has been expanded by the following initial sentences (see lines 458-464, revised version): “Under cardiovascular disease conditions, such as IR and HF, multiple pathways of MQC become exhausted, thus exacerbating cell death and cardiac dysfunction. In this scenario, a therapeutic strategy aimed at enhancing the multilayer process of MQC is expected to improve energy supply leading to a better preservation of contractile force and inhibition of the fibrosis-cell death axis. The available data indicate that restoration of euthyroidism restoration, in the setting of post IR LT3S, increases myocardial resilience to stress stimuli in association to the retrieval of preventive and reparative pathways that are relevant to cardiomyocytes mitochondrial bioenergetics and QC”.

In addition, we added the following final comment (see lines 497-501): “In conclusion, increasing experimental data indicate the mitochondrion as a main target of TH-mediated cardioprotection. An approach that aims to euthyroidism restoration, by restoring the free T3 plasma levels without major systemic alterations, may represent the best strategy to limit the evolution of HF while avoiding the undesirable hypermetabolic and pro-oxidant effects of higher TH doses”.

Moreover, to address each and every point raised by both the reviewers, other sections have also been amended/broadened as reported in the specific point-by-point rebuttal.

Special comments:

1)  Title: The title is not appropriate since the review is focused on the effects of reconstitution of euthyroidism. The title is too general for the content of the review.

R: Accordingly the title has been modified as follows : Protective effects of euthyroidism restoration on mitochondria function and quality control in cardiac pathophysiology”.

·      Introduction:

2)       Physiological functions of TH should be included 

3)      Molecular signaling of TH in the cardiovascular system (e.g. THRa and THRb?)

4)       The specialization on euthyroidism should be introduced/explained. Why is hyperthyroidism not discussed?

5)       Mention/Discuss local versus systemic effects of TH.

R: We thank the reviewer for these insightful suggestions. To address the above points (from point 2 to point 5), the following sentences have been added in the introduction section (see lines 69-81 revised version): “The crucial involvement of TH signaling in the regulation of cardiovascular physiology has been clearly established in the last decades (17-18). TH enhance cardiac performance and heart rate both locally, by influencing myocardial gene expression or protein activity, and indirectly (systemically), by decreasing the peripheral resistance. One predominant mechanism by which the biologically active TH, triiodothyronine (T3), directly affects heart function is by binding to TH nuclear receptor alpha and beta isoforms (THRa and THRb). Besides this so called genomic action of TH, non-genomic TH activities exist that are not dependent on THR binding and are initiated at plasma membrane level (19). Since all these modality of TH signaling are present in the cardiovascular system, it is not surprising that maintenance of euthyroidism is of crucial importance to preserve cardiac performance, as documented by the opposite cardiovascular alterations observed under hypo and hyperthyroidism (18). Besides overt thyroid dysfunctions, even milder alterations of TH homeostasis may adversely impact on heart function (18)….”.

·      Discussion:

6)       Direct and indirect effect and lack of knowledge with regard to direct target of T3/T4 should be discussed.

R: To argue this topic the following sentence have been inserted in the conclusion section (see lines 482-485 revised version): “So far the vast majority of the investigations aimed at identifying TH targets exploited indirect approaches, mainly based on comparative analysis of RNA or protein abundance (121). Further studies on myocardial chromating occupancy by liganded and unliganded THR are necessary to clearly distinguish between direct and indirect influence of TH on cardiac gene transcription.”

7)       What issues be considered to reconstitute euthyroidism

R: We have addressed this topic in the conclusion section by adding the following sentences (see lines 486-494, revised version): .., the available data on T3 cardiac effects should provide a helpful guidance to optimize the benefit to risk profile when considering LT3S correction. To be effective against adverse remodeling, restoration of euthyroidism, should be performed in the early phase of the post IR wound healing process, a timing when the activation of noxious mitochondrial regulatory networks have not yet produced permanent effects and may be prevented or blunted by T3 replacement. Accordingly, TH administered early after IR enhanced cardiac function and prevented left ventricle remodeling (27, 30-31 66). By contrast, T3 treatment started one week after IR improved heart performance without reversing cardiac remodeling, which may be the result of systemic rather than local myocardial effects (122).”

8)       Sex-related differences are observed in the clinic. This impact should be discussed briefly

R: the topic has been briefly mentioned in the discussion with the addition of the following paragraph (see lines 494-496, revised version): “Finally sex-specific differences in the long term cardiac responsiveness to T3 have been reported in post acute myocardial infarction patients, which should be taken into account for a better personalization of a TH replacement strategy (123).”

9)      Figure 1: Role of T4 was not mentioned anywhere

R: Accordingly, the role of T4  has been mentioned in the caption of fig.1 as follows (see lines 117-118, revised version): “T3 and T4 decrease reactive oxygen species (ROS) production by multiple mechanisms;…”

10     Please, check if all abbreviations are explained (e.g. NLPR3 in fig legend 3; THR in fig legend 1,….)

R: Accordingly all figures were checked and the missing explanations were added.

11      Reference 35 is not included in the text.

R: The reference list and consequently their citation in the text have been amended and new references have been incorporated in the revised manuscript

Reviewer 2 Report

1)    The title, “Direct and indirect effects of thyroid hormones on mitochondria function and quality control in cardiac 3 pathophysiology”, indicated the review would focus on mitochondrial function and mitochondrial quality control, but few statements to be mentioned regarding mitochondrial quality;

2)    Many articles talked about the relationships between TH and cardiac mitochondrial replication, mitochondrial biogenesis, mitochondrial fusion/fission. It’s necessary to review the roles of TH in above;

3)    The influences of TH on cardiac mitochondrial function are associated with mitochondrial DNA- and nuclear-encoded mitochondrial gene expression as well as their protein synthesis via TH-thyroid hormone receptor binding and the binding-induced pathways involving in, please mention the comment.

4)    Some language problems:

Ø  Line 313, what is the mDNA?

Ø  Line, Pink1stabilization should be Pink1-stabilization;

Ø  Be careful to standard manuscript format following journal’s introduction before submitting your manuscript.

Author Response

Reviewer #2 (Comments to the authors)

We are grateful to the reviewer for his/her constructive comments and suggestions.

1)    The title, “Direct and indirect effects of thyroid hormones on mitochondria function and quality control in cardiac pathophysiology”, indicated the review would focus on mitochondrial function and mitochondrial quality control, but few statements to be mentioned regarding mitochondrial quality;

R:  In reviewing the mitochondrial effects of euthyroidism restoration we dedicated 3 entire sections to mechanisms of mitochondrial quality regulation that are: 1) mitochondrial repair systems ; 2) mitochondrial clearance (mitophagy); and  3) mitochondrial protein import. To be more focused on the review content, the title has been changed as follows: “Protective effects of euthyroidism restoration on mitochondria function and quality control in cardiac pathophysiology” see also the replay to comment 1) of  rev. #1.

Moreover, as suggested by the reviewer, the issue of mitochondrial quality has been better explained and underscored in the introduction section as follows: “To preserve high mitochondrial quality and energy reserve, the cell needs to integrate, near concurrently, multiple mechanisms of mitochondrial quality control (MQC) consisting in recognition and isolation of irreparably damaged mitochondrial components, their targeting to the clearance systems and their replacement with plenty functional ones via mitochondrial biogenesis. When this quality check processes are impaired, as in cardiac disorders, mitochondria become more susceptible to danger signals which predispose to energy crisis and disease progression (1-2).” (see lines 36-42, revised version).

2)    Many articles talked about the relationships between TH and cardiac mitochondrial replication, mitochondrial biogenesis;, mitochondrial fusion/fission. It’s necessary to review the roles of TH in above

R:  Accordingly, new references have been incorporated and the last paragraph has been extended as follows (see lines 415-427): “Most of the mitochondrial proteome is encoded by the nuclear genome, while the mitochondrial DNA encodes for 13 essential polypeptides of the ETC complexes. A tight coordination of the two genomes is necessary to ensure the proper assembly of all the mitochondrial components. The master regulator of the nuclear mitochondrial crosstalk is PGC-1a, the co-activator that interacts with a variety of transition factors involved in mitochondrial biogenesis, bioenergetics and antioxidant activity (108). Among the PGC-1a targets, the nuclear-encoded mitochondrial transcription factor A (Tfam) is responsible for the replication and repair of the mitochondrial genome and for the transcription of  mtDNA encoded genes (109).”

In this review we have chosen to focus on mitochondrial import machinery since the role of TH in the regulation of Pgc1a and Tfam has been extensively discussed in previous manuscripts, some of which are mentioned in the following sentence of the new version for the benefit of interested readers to this specific topic (see lines 425-427, revised version): “The key role of TH in the up-regulation of PGC-1a and Tfam, has yet been described in previous reviews (27, 110-114), here we focus on the regulation of the mitochondrial protein import system by T3.” 

Unfortunately, we did not find any article dealing with the role of TH in the regulation of cardiac mitochondrial fusion or fission. To the best of our knowledge, the only available findings are the mRNA expression data relative to Mfn1 and Mfn2 obtained in our laboratory (see ref 16).  However, following the suggestion of the reviewer, few information on T3 dependent regulation of mitochondrial fusion and Mfn1 have been added in the revised manuscript: (see lines 376-379, revised version): “In a system biology study, a post IR LT3S was associated to impaired mitochondrial activity in association to reduce expression of key markers of mitochondrial biogenesis, mitophagy and fusion including Pgc1a, Bnip3, Mfn1 and Mfn2 (16). These alterations were mitigated by early and short term T3 replacement”.

3)    The influences of TH on cardiac mitochondrial function are associated with mitochondrial DNA- and nuclear-encoded mitochondrial gene expression as well as their protein synthesis via TH-thyroid hormone receptor binding and the binding-induced pathways involving in, please mention the comment.

R:  Accordingly, the reviewer’s comment has been added (see lines 421-424, revised version) : “Several lines of evidence have established a clear connection between TH induction of mitochondrial activity and up-regulation of the entire process of mitochondrial biogenesis (110-114). One main mechanism is via T3 binding to TH receptors located in both the nuclear and mitochondrial compartments (110-114).”

Some language problems:

4) Ø  Line 313, what is the mDNA?

R:  Sorry for the mistake, the typing error has been amended as: mtDNA, (see line 361 revised version).

5) Ø  Line, Pink1stabilization should be Pink1-stabilization;

            R:  The correction has been made in the revised manuscript.

6) Ø  Be careful to standard manuscript format following journal’s introduction before submitting your manuscript.

R:  :  We checked that the format of the manuscript complies with the instructions provided by the journal.